# Development of Self-Assembly Methods on Quantum Dots

**DOI:** 10.3390/ma16031317

**Published:** 2023-02-03

**Authors:** Qun Hao, Hongyu Lv, Haifei Ma, Xin Tang, Menglu Chen

**Affiliations:** 1School of Optics and Photonics, Beijing Institute of Technology, Beijing 100081, China; 2Beijing Key Laboratory for Precision Optoelectronic Measurement Instrument and Technology, Beijing 100081, China; 3Yangtze Delta Region Academy of Beijing Institute of Technology, Jiaxing 314019, China

**Keywords:** quantum dots, self-assembly, methods

## Abstract

Quantum dot materials, with their unique photophysical properties, are promising zero-dimensional materials for encryption, display, solar cells, and biomedical applications. However, due to the large surface to volume ratio, they face the challenge of chemical instability and low carrier transport efficiency, which have greatly limited their reliability and utility. In light of the current development bottleneck of quantum dot materials, the chemical stability and physical properties can be effectively improved by the self-assembly method. This review will discuss the research progress of the self-assembly methods of quantum dots and analyze the advantages and disadvantages of those self-assembly methods. Furthermore, the scientific challenges and improvement in the self-assembly method of quantum dots are prospected.

## 1. Introduction

Semiconductor inorganic nanocrystals with a radius smaller than Bohr excitons are called quantum dots (QDs). In the past few decades, QDs have been the subject of intense interest in physics, materials science, and electrical engineering. The application of QDs for optoelectronic devices has been applied in the lighting, display, and biomedical fields [1,2,3] due to their unique optical properties such as tunable absorption and emission wavelengths [4], ultra-narrow half-width [5], and high luminescent quantum yield [6]. At present, semiconductor QDs mainly contain: II–VI group QDs, III–V QDs, and halide perovskite. The II–VI group QDs normally show excellent luminescent character [7]. However, II–VI group QDs contain heavy metal ions, so they are generally toxic. Compared with the II–VI group QDs, III–V group QDs are less toxic. However, their optical characteristic is worse than the II–VI group QDs [8]. Therefore, researchers have carried out research on the optimization of the optical performance of III–V group QDs [9,10]. Recently, halide perovskite QDs have aroused great interest given its simple synthesis and high luminescence yield, and it has become an important category [11,12,13]. However, the chemical stability of halide perovskite QDs is severely limited by external environmental factors. Although a series of solutions such as surface defect modification and ligand passivation have been proposed for the chemical stability of perovskite QDs, the optimization effect is still very limited [14]. Therefore, improving the chemical stability of halide perovskite QDs is still an important research topic.

The self-assembly method could effectively solve the bottleneck of the above kinds of QDs, as shown in Figure 1. Self-assembly systems mean that molecules, atoms, and ions interact with each other by non-covalent (hydrogen bonds, van der Waals bonds, and ionic bonds) interactions to form larger structures. Compared with manual assembly, the advantages of self-assembly include less defects, high order, and adjustable size. Reversible self-assembly of nanoparticles into ordered structures is essential for both fundamental study and practical applications [15].

The self-assembly of QDs occurs due to the relaxation of strain energy in epitaxial systems in which the deposited material has a lattice parameter that is significantly larger than that of the underlying material [16]. The self-assembly method can effectively improve the stability and mobility of QDs. Moreover, the self-assembly method can suppress the interference of heat flow and non-radiative transitions on QDs. In addition, the QDs have an adjustable size after self-assembly, which further expands the application prospect of QDs in real-time biosensing, environmental monitoring, and in the biomedical field [17,18,19]. Furthermore, the QDs of different types and their assemblies are of immense utility for the photonic crystal-coupled emission (PCCE) research community [20]. Nowadays, researchers have observed the phenomenon of self-assembly in a variety of QD materials. In 2009, García-Ruiz et al. discovered that the precipitation of barium or strontium carbonates in alkaline silica-rich environments leads to crystalline aggregates. These aggregates belong to pure inorganic self-assembled nanocrystalline materials [21]. In 2019, Grünwald et al. used molecular dynamics computer simulations to study the self-assembly of these nanocrystals over a broad range of ligand lengths and solvent conditions, as shown in Figure 2a [22]. The results showed that small differences in nanoparticle shape, ligand length and coverage, and solvent conditions can lead to markedly different self-assembled superstructures due to subtle changes in the free energetics of ligand interactions. In 2021, Xing et al. reported a unique irradiation-triggered self-assembly and recrystallization phenomenon of crystalline carbon nitride QDs terminated by abundant oxygen-containing groups, which open up new possibilities of manipulating carbon nitride nanomaterials via controlled assembly, as shown in Figure 2b [23]. 

Although extensive work has been conducted, the demand for simple, reversible, and versatile self-assembly methods is still a central issue in current nanoscience and nanotechnology. Due to the different types of QDs, their structures and properties show significant differences. Hence, the variety of QDs needs to be self-assembled by different methods. The current mainstream self-assembly methods of QDs include the molecular beam epitaxy method, electrochemical method, chemical vapor deposition method, and ligand exchange method. In 2019, Hou et al. introduced a method of self-organizing synthesis and the coupling of QDs to directly generate quantum chains (QCs) [24]. Using oleamine and oleic acid as structure directing agents, the QDs were interdigitated into microscale chainlike superchain crystals. This growth method is quite different from other reported self-assembly methods. This method does not introduce new surfactant ligands or solvent evaporation, whereas the QC superstructure can be generated directly from the colloidal growth process. At the same time, this growth strategy can be extended to the growth of other metallic chalcogenide compounds such as indium copper ternary (CuInS_2_). Additionally, these chain-like structures have been proven to be easily adapted for practical applications such as lithium-ion batteries. QC-activated LIB cells exhibit considerably higher electrochemical Li^+^ storage capacity and charge mobility than discrete QDs and bulks due to extremely small and minimum lattice stresses at adjacent crystal boundaries. In 2022, based on the self-assembly of carbon QDs, Gao et al. introduced an innovative method for preparing nanomaterials under the action of a metal catalyst [25]. Carbon QDs were synthesized by the one step hydrothermal method using citric acid as the carbon source, ethylenediamine as the passivator, and FeSO_4_·7H_2_O as the precatalyst. Under the action of a gas–liquid interface template and the metal catalyst, carbon QDs were self-assembled to form a thin film. The film exhibited excellent photoluminescence properties and holds great potential as a high-performance electrode material for supercapacitors. In 2022, Rai et al. designed and developed a simple and rapid method of adiabatic cooling. Precise nanocomponents with adjustable optical and morphological properties were obtained by cooling the original nanoparticle solution to different temperatures and time intervals. This research provides a new idea for the assembly of QDs [26]. In addition, Aftenieva et al. found that QDs could be self-assembled by cost-effective wet chemical methods in 2023 [27]. This method cannot be restricted by the preferential substrate growth in a traditional semiconductor laser. The synthetic colloidal QDs improved the optical stability and optical gain, bringing QD-based lasers closer to practical applications and circuit integration. In addition, additional flexibility is provided in terms of the physical and chemical properties.

However, these methods still have their own limitations, which hinder the further optimization of QD performance. Here, we summarize the common problems existing in the current self-assembly methods of QDs and we analyze the self-assembly methods suitable for various QDs in the review. This review puts this information together to provide the self-assembly methods for successfully growing QDs for some of the most commonly used semiconductor material systems, and is expected to be useful to both novice and experienced researchers of QD self-assembly. In addition, this review is helpful for researchers to quickly find suitable self-assembly methods for different QD materials.

In this review, we intend to introduce the current self-assembly methods on QDs. Moreover, the latest research progress and limitations of self-assembly methods on QDs are described and evaluated in detail. We hope that such a comprehensive review will provide extensive guidance for the development of self-assembly methods on QDs and the design of high-quality QDs.

## 2. Self-Assembly Methods of Quantum Dots

### 2.1. Molecular Beam Epitaxy Method (MBE)

The development of MBE technology has promoted the development of semiconductor QDs to a great extent, and is the most important technology for the preparation of semiconductor materials (thin films and heterojunctions). The epitaxy growth process is generally the accumulation of atoms on the surface of the solid substrate, and then these atoms are adsorbed and migrated on the surface of the substrate. Finally, some atoms nucleate on the surface of the substrate, and the epitaxy grows into a new layer of material [28]. Due to the great difference between the epitaxial layer material and substrate material in the lattice constant, interface energy, surface energy, and other parameters, epitaxial growth mainly has the following three modes:(1)Frank van der Merve mode

When the interaction between the epitaxial layer molecules and the atoms on the substrate surface is stronger than that between the epitaxial layer molecules, the epitaxial molecules will first interact with the atoms on the substrate surface. The epitaxial molecule will first form some discrete atom-monolayer high two-dimensional crystal nuclei on the substrate, continue to grow, and then form a complete crystal plane. With the increase in the epitaxial material deposition, the epitaxial material will continuously form two-dimensional crystal nuclei on the newly formed complete crystal plane, and grow layer by layer in this way, finally forming a thin film. The growth diagram is shown in Figure 3a.

(2)Volmer–Weber mode

When the interaction between the epitaxial layer molecules and the atoms on the substrate surface is stronger than that between the epitaxial layer molecules, the epitaxial molecules will first interact with the atoms on the substrate surface [29]. The epitaxial molecule will first form some discrete atom-monolayer high two-dimensional crystal nuclei on the substrate, continue to grow, and then form a complete crystal plane. With the increase in the epitaxial material deposition, the epitaxial material will continuously form two-dimensional crystal nuclei on the newly formed complete crystal plane, and grow layer by layer in this way, finally forming a thin film. The growth diagram is shown in Figure 3b.

(3)Stranski–Krastanow mode

This growth pattern was discovered by Stranski et al. in 1958 [30]. If the lattice mismatch between the epitaxial material and the substrate material is between 5% and 10%, two-dimensional layered growth will be performed first, and once some critical thickness is reached, a spontaneous transition to 3D island formation occurs. The self-assembled growth of semiconductor QDs is obtained by using the S–K growth mode. The growth diagram is shown in Figure 3c.

The above three modes are compared in Table 1.

By the MBE technique, epitaxy layers of different thicknesses can grow directly on the substrate. The principle is to place substrate and multiple evaporator sources in the system of an ultrahigh vacuum environment. Figure 3d shows that when we heat different evaporator sources, the material in the evaporator source is sprayed onto the substrate in the form of molecular beams in a certain proportion, and finally forms an epitaxial film on the surface of the substrate. Figure 3e shows a transmission electron micrograph of a cross sectional view of a GaAs/Al_x_Ga_1−x_As superlattice [31].


Figure 3(**a**–**c**) Three main modes of film growth. (**a**) Frank van der Merve mode. (**b**) Volmer–Weber mode. (**c**) Stranski–Krastanow mode. (**d**) Top view of a simple MBE chamber showing the essential growth sources, shutters, beam flux detector, and the RHEED system for monitoring structure during growth [32]. Copyright 2002, *Surface Science*. (**e**) Scanning electron micrograph of the cross-sectional view of alternating layers of GaAs (dark lines) and Al_x_Ga_1−x_As (light lines) grown by MBE [31]. Copyright 1999, *Journal of Crystal Growth*.
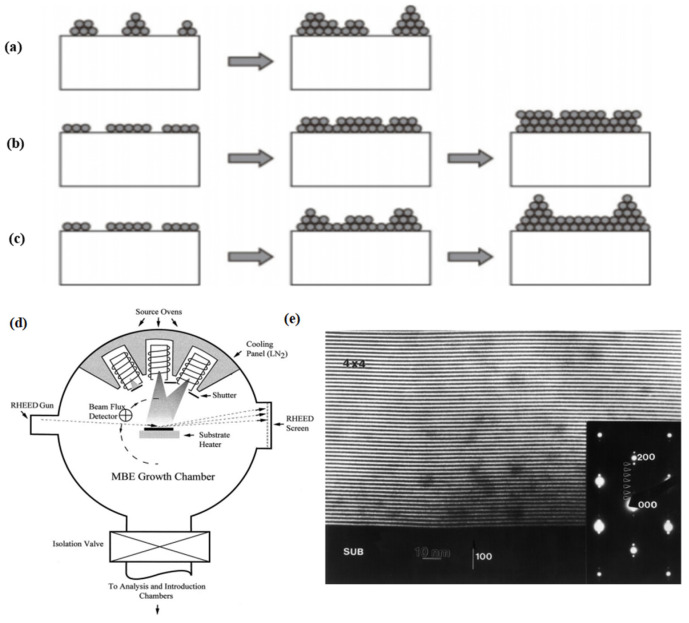



In 1968, LePore et al. found that the epitaxial growth of GaAs and GaP could be obtained at about 500 °C while studying the reflection of pulsed molecular beams of Ga and As_4_ on the surface of GaAs in ultrahigh vacuum. In 1969, Cho published an experimental result of great significance. He reported the first in situ observations of the MBE growth process using high energy electron diffraction. This structural analysis capability has proven to be critical for characterizing MBE epitaxy, as it provides immediate feedback on the influence of growth conditions on the structure of the films. Cho demonstrated that MBE growth could produce flattened ordered layers of atoms; as a result, these studies marked the beginning of the use of MBE for device manufacturing [32,33]. The researchers then used this new technique—MBE—to synthesize the first QDs with a one-dimensional quantum constraint. However, going further, the production of a nanostructure capable of providing two- or three-dimensional quantum confinement was a huge technical challenge at the time. By the early 1990s, researchers had developed two techniques for implementing three-dimensional constraints. In 1995, Saitoh et al. carried out the optical characterization of InAs QDs prepared by the MBE method [34]. They found that the photoluminescence peak energy through the self-assembled InAs QDs was unexpectedly high. According to these optical characterizations, the intrinsic strain distribution formed during the crystal growth plays an important role in the analysis of luminescence for self-assembled InAs QDs. Further analysis is required to verify the intrinsic strain and piezo electric effect in the self-assembly QDs. In 2001, Matsumura et al. fabricated a ZnSe diode with a CdSe multi QD layer by the MBE technique and observed a green emission when a current was injected into the diode [35,36]. The structure is shown in Figure 4a. Figure 4b,c shows the luminescence spectra of the CdSe and diode, respectively. In 2007, Ohkuno et al. established the optimal growth conditions for CdSe QDs by changing the growth parameters such as the supply rate of the gallium arsenide substrate, growth interruption time, and temperature [37]. Figure 4d shows the dependence of the photoluminescence spectra on the rate of the CdSe source supply. Figure 4e,f shows the dependence of the photoluminescence spectra on the growth interruption time at a source supply rate of 0.01 and 0.1 ML/s, respectively. Then, the influence of strain force on the self-assembly of QDs was studied in depth by Sautter et al. in 2020 [38]. They studied the effects of compressive and tensile strain force on the self-assembled QDs. This study lays the foundation to create a new type of self-assembled QD with unique properties for future applications. In 2023, Vallejo et al. demonstrated the self-assembled growth of In_0.5_Ga_0.5_As QDs on GaSb(111)A surfaces by MBE. QDs with good structure and optical quality were generated by MBE. The deposition of In_0.5_Ga_0.5_As onto the GaSb(111)A surfaces resulted in the self-assembly of QDs under a 4% tensile strain. They found that the InGaAs(111)A QDs were fully strained and free from defects below some threshold size. Due to the residual tensile strain, the InGaAs(Sb)/GaSb(111)A QDs realized the tunable emission between mid-infrared bands [39].

The advantages of MBE growth are as follows: first, there is almost no impurity in the epitaxial layer, because the whole growth process is carried out in a clean ultra-vacuum environment; at the same time, the ejected molecular beam will not collide with the impurity molecules and scatter, so the epitaxial layer has high purity. Second, the growth rate is slow during the growth process, so the thickness and structure of the epitaxial layer can be precisely controlled. Third, during the growth process, the bonding energy of the materials is decreased under the ultrahigh vacuum environment, and the film forming temperature is decreased, which can reduce the thermal defects and the diffusion of impurities in the substrate and epitaxial layer. As a result, the film is relatively smooth and flat.

However, at the same time, this technology also has some defects: first, the growth rate is slow, so the growth process takes a long time. Second, the material growth needs to be carried out in ultrahigh vacuum (*p* ≤ 10^−9^ torr), and the growth environment is quite strict. Third, the substrate can only grow a material that has low degrees of lattice mismatch with the substrate. When the lattice mismatch between the substrate and material is high, the thickness of the epitaxial layer will be uneven. Fourth, this technology cannot carry out batch production. 

Based on the shortcomings of molecular beam epitaxy, researchers have proposed other improved methods successively such as gas phase epitaxy, liquid phase epitaxy, atomic layer epitaxy, and laser molecular epitaxy, etc. The growth of QDs by droplet epitaxy technology is not limited by the lattice constants of the substrate and epitaxy materials, nor does it need to be carried out at higher temperatures, so the operation is simple and easy to realize. The droplet epitaxy technology can not only easily control the density, size, and distribution of QDs by changing the substrate temperature, the ratio of epitaxy materials, deposition amounts, and other parameters, but also forms special structures such as quantum pairs and quantum rings. Of course, these methods have their own advantages and disadvantages and still need to be improved.

### 2.2. Gravity Sedimentation Method

The gravity sedimentation method under the action of a gravity field is a method to realize colloid self-assembly by applying the field force. The principle is that colloidal particles are dispersed in the solvent, the colloidal particles settle slowly under the action of the gravity field, and self-assemble to grow on the substrate. The key to realizing the colloid self-assembly by gravity lies in the control of several important parameters such as the colloid particle size and settling velocity. The descending process of particles obeys Stokes’ law, and the settling velocity is closely related to the particle density, size, and the viscosity of the dispersion medium. For particles of larger size, the sedimentation rate is too fast, which can easily cause the uneven thickness of the sample. However, for particles of smaller size, the sedimentation rate is very slow, the sedimentation process often lasts for several months, and it may even be difficult to obtain the deposition sample. 

Compared with the early self-assembly method by gravity sedimentation, the vertical sedimentation method driven by capillary force has been found up to now. The vertical sedimentation method was first proposed by Nagayam in 1995 [40]. The principle is to assemble colloidal particles in an orderly manner by the interaction of capillary force and surface tension. They found that as the particles spread out over the substrate surface, their self-assembly would suddenly start by removing the solvent (such as evaporation), causing the liquid layer to thin out and eventually form a hexagonal lattice. In addition, with the increase in solvent concentration, the phenomenon becomes more obvious. As a result, they proposed that the particles could be induced to self-assemble in the liquid layer by properly controlling the thickness of the liquid layer. They delved deeper into the phenomenon that the self-assembly of particles could be induced by preparing stable liquid films on suitable surfaces whose thickness is equivalent to the geometry of the particles. At the same time, the equilibrium thickness of the liquid film is obtained by the force balance between the osmotic pressure (separation pressure) in the wet film and the capillary pressure generated by a crescent-like surface formed on the wet wall of a container. This method can produce high quality samples with fewer defects and a more uniform material thickness. 

This method was later developed by Colvin et al.; inserting the wetted substrate vertically into the colloidal particle solution will form a meniscus area on the substrate, where the solvent at the surface will constantly evaporate and the surrounding solution will constantly flow in. The colloidal particles are driven into a growth zone, and the particles are closely arranged under the capillary force of the interaction between the colloidal particles [41]. The thickness of the film can be changed from a single layer to hundreds of layers by controlling the concentration of the colloid solution and the size of the particles. There are three limitations to this method: first, it takes a lot of time for the solution to evaporate, so the preparation time of the film is long. Second, the evaporation of the solvent will cause the concentration of the solution to change, which will change the thickness of the film along the growth direction. Third, the deposition is limited to smaller colloidal particles, and the deposition rate of particles is slower than the evaporation rate of the solvent. For larger colloidal particles, the sedimentation rate of particles is higher than the evaporation rate of the solvent, so the film thickness cannot be accurately controlled. 

To solve the problem of large particles settling too quickly, Norris et al. reduced the influence of gravity on particle deposition by creating a temperature gradient on the colloidal particle solution to generate continuous convection [42]. Ozin et al. proposed a method of isothermal heating evaporation-induced self-assembly, which is a fast, generally reliable, and repeatable method [43]. It can be used to produce colloidal crystal films with large area, high quality, and ordered arrangement. In this method, the prepared films are not limited by particle size. The experimental device is shown in Figure 5a–d. This method has been proven to be able to obtain large-area, high-quality, and ordered colloidal crystal films. Scanning electron microscopy images of the prepared films of different thicknesses is shown in Figure 5e,f. When the substrate is placed at different angles, the resulting membrane imaging is shown in Figure 5g,h [44]. However, the temperature conditions should be strictly controlled, and the temperature deviation should not exceed 0.05 °C. Then, Holgado et al. developed self-assembly under an applied electric field, and proposed the use of the electric properties of colloidal particles to induce the self-assembly [45]. The method solves the problem that the colloid particles with too large or too small size are difficult to assemble. The principle of the electric field self-assembly method is to use the magnitude and direction of an applied electric field to induce charged colloidal particles to accelerate (small particles) or decelerate (large particles). Thus, the problem of large particles settling too fast and small particles that are not easy to settle is solved. Here, the advantage of using electric field force induction is that the electric field can produce non-contact force and this non-contact force can cover the whole space, preventing the thickness of sample from being uneven. The preparation method has a short time period and is not limited by the size of particles, but the charge density on the surface of colloidal particles needs to be strictly controlled.

In 2018, Mukai et al. improved the properties of solar cells by using superlattices of PbS QDs prepared by the sedimentation method [46]. Controlling the vapor pressure of the atmosphere can change the evaporation rate of the solvent, and the authors used this method to adjust the settlement time of the QDs. As the deposition process continues, the PbS QD grains grow into a large superlattice. A kind of solar cell is prepared by using the QD superlattice film as the absorption layer. Compared with traditional solar cells, the short circuit current density and power conversion efficiency of solar cells prepared by this method were improved. At the same time, they found that the short-circuit current density of the solar cells doubled when the deposition was slow. This shows that the settlement time affects the uniformity of the interface. 

By using a template, they deposited PbS QDs onto a microporous array template to form a superlattice thin film [47]. As shown in Figure 6, it was found that the higher-order emission lifetime on the template was more than twice that on the planar substrate. The lifetime of the QD superlattice films prepared by the deposition method was long, which shows that the template widened the grain size of the single QD superlattice.

For the gravity sedimentation method, it has the following advantages: first, the method for preparing thin films is simple, the experimental environment and requirements for experimental equipment is low and is easy to meet. Second, the thickness of the prepared sample can be controlled. However, because colloidal particles settle under the gravitational field, the sedimentation rate of particles of different sizes is different, which leads to the uneven thickness of the prepared sample. Meanwhile, the whole preparation process takes a long time, which is not conducive to large-scale production.

### 2.3. Colloidal Solution Method

The colloidal solution method is based on the self-assembly characteristics of the colloid, and nanoclusters are easily dispersed in the solvent to form a colloidal solution. Therefore, as long as the appropriate conditions are available, the nanoclusters can be assembled to form a regular arrangement. In Figure 7a, the self-assembled superstructures are usually named supra QDs (SQDs), which are typically composed of hundreds of a-few-nanometer-sized QDs three-dimensionally assembled by an oriented attachment. The size of the SQDs can be tuned from tens of nanometers to over a hundred nanometers. Furthermore, the collective properties of the colloidal SQD structure are not only controlled by the properties of nanoparticles, but also by the symmetry, orientation, phase, and size of the superstructure. This combination of properties makes colloidal superstructures that are highly susceptible to remote stimuli or local environmental changes, providing a prominent platform for the development of stimuli-responsive materials and smart devices. At present, the colloidal solution method has been successfully used for the self-assembly of various metal nanoparticles and some semiconductor QDs. 

In 2009, Zhang et al. successfully realized the self-assembly on ZnO QD/functionalized multi-walled carbon nanotube composite by the colloidal solution method [48]. In this method, the functionalized multi-walled carbon nanotubes were added into the QDs of the ZnO alcogels. Under the ultrasonic condition or being magnetically stirred, ZnO QDs and functionalized multi-walled carbon nanotubes can self-assemble to form a composite material. In Figure 7b,c, it can be clearly observed from the electron microscopy image that monodispersed ZnO QDs were anchored stably on the functionalized multi-walled carbon nanotubes. Compared to the raw multi-walled carbon nanotubes with metallic characteristics, the composite materials exhibited the characteristics of the semiconductor clearly after self-assembly.

In 2012, Singh et al. reported that nanosecond pulsed laser ablation of a zinc rod placed on the bottom of a glass vessel containing methanol was used to produce a colloidal solution of drop shaped ZnO QDs [49]. The results indicate that ZnO QDs can self-assemble into various dendritic nanostructures in a colloidal solution, as shown in Figure 7d–f. It can be seen from the electron microscope observation that the self-assembly ZnO QDs exhibited linear axis symmetrical branching, linear axis asymmetrical branching, and curvilinear axis asymmetrical branching. Moreover, most of the ZnO QD self-assembled structures showed linear growth. Under 325 nm excitation, the photoluminescence spectrum of the self-assembled ZnO QDs covered the range from 350 to 600 nm. UV bands may be the consequences of free excitonic transition and bound excitonic transition. Blue emission is the consequence of the electronic transition from the extended zinc interstitial state to the valance band. Oxygen interstitial, oxygen vacancies, and antisite oxygen trigger the green emission.

In 2016, Park et al. reported the research of the stepwise self-assembly of cadmium chalcogenide QDs to colloidal SQDs. In the case of CdSe SQDs, zinc-blende seeds (primary QDs) act as the building block for the formation of the 3D assembled structures with discrete intermediate nanostructures [50]. The 4 nm tetrahedral shaped QD seeds assembled into a large tetrahedron of 20 nm. The 20 nm tetrahedrons self-assembled into a larger tetrahedron of 40 nm, as shown in Figure 7g. The discrete-in-size and sequential assemblies were followed by conventional growth from the remaining precursors and ripening within the particles to result in spheroidal SQDs.

Colloidal solution self-assembly methods are also important for optimizing the performance of QDs. At present, several research studies have previously reported the construction of 2D assemblies from colloidal nanostructures in solution. In 2017, Park et al. fabricated a closely packed graphene QD film from colloidal solutions using a self-assembled method, and investigated the optical and electrical characteristics of the heteroaccumulation graphene/graphene QD film structures [51]. In the self-assembled process, destabilized charged particles are trapped at the liquid/liquid or liquid/air interface. Since graphene QDs are negatively charged, graphene QDs in solution can gradually be transformed into continuous solid films. In Figure 8a, the graphene covering both the graphene QD film and SiO_2_ regions shows a dual Dirac point. This behavior demonstrates the role of a graphene QD film as a buffer layer that isolates graphene from undesired p-doping. As shown in Figure 8b, the carrier mobility of graphene QDs remained constant regardless of the doping densities, which resulted in strong n-type doping without a decrease in the charge carrier mobility.

In the same year, researchers designed and biosynthesized artificial polypeptides PC10ARGD@QDs as vehicles for encapsulating hydrophobic materials [52]. In this method, the hydrophobic CdSe@ZnS QDs were loaded into the polypeptide nanogels by ultrasonic treatment. The PC10ARGD@QDs hybrid nanogels by colloidal solution self-assembly showed good fluorescence stability and excellent biocompatibility. Figure 8c shows that the PC10ARGD@QDs hybrid hydrogels exhibited a fluorescence emission under different pH values. The emission intensity of the PC10ARGD@QDs hybrid hydrogels at pH 3 remained at 60% of intensity at pH 9. Moreover, the viabilities of the PC10ARGD@QDs hybrid hydrogels were still above 80% at the QD concentration of 2.5 nm. In addition, the fluorescence intensity of the HeLa cells incubated with the PC10ARGD@QDs hybrid nanogels was much higher than that of the without self-assembly PC10ARGD polypeptide, as shown in Figure 8d. These results demonstrate that PC10ARGD@ QDs as nanocarriers are expected to have great potential applications in biomedicine.

Due to the weak recognition of molecules in the assembly process, such assembly processes are difficult to control, and the requirements for the assembly conditions are very strict. The regular three-dimensional superlattice can be assembled by the colloidal solution method, which is difficult to achieve by other methods. However, colloidal solution methods are typically not suitable for the fabrication of assemblies over large areas or in 3-D accessible formats. Moreover, these self-assembly methods are sensitive to defects and considerable room remains to improve their electrical transport properties. 

### 2.4. Electrochemical Method

Electrochemistry is a powerful technique. It provides the measurement of the Fermi level and reversible electrochemistry allows for the absolute measurements of the filled and empty state energies with the application of a voltage. Electrochemistry can also be used to learn about chemical processes and decomposition at QD surfaces. Self-assembly of nanostructures using electrochemical means is now a well-established technique. As shown in Figure 9a, the electrochemical self-assembly method is mainly divided into the following steps. The QD solution was placed in a three-electrode system (Pt wire as the working electrode, carbon electrode as the counter electrode, Ag/AgCl/sat. KCl as the reference electrode). The ligand on the surface of the QDs was removed by applying positive voltage oxidation on the Pt wire. Then, due to the oxidation of the surface of QDs, the aggregation of QDs was induced. The adjacent QDs particles were connected together, and finally self-assembled to form a controllable nanostructure.

In 1999, Bandyopadhyay et al. reported some results pertaining to the magnetic, optical, and topographic properties of electrochemically self-assembled QDs [53]. The results indicate that the CdS QDs self-assembled by the electrochemical method showed a two-dimensional hexagonal arrangement. This self-assembled structures have important applications in magnetics, electronics, and nonlinear optics. In 2000, Balandin et al. further researched the CdS QD quasiperiodic arrays with a characteristic feature size of 10 nm by the electrochemically self-assembly method [54]. The QDs were synthesized using the electrochemical deposition of CdS into a porous anodized alumina film. The results suggest that QDs self-assembled by the electrochemical method have the advantage of being inexpensive and robust. Moreover, the QD arrays are suitable for infrared photodetector applications. In 2002, Kouklin et al. reported that cylindrical QDs with a diameter of 8 nm and height of 3–10 nm, and wires with a diameter of 50 nm and height 500–1000 nm, could be self-assembled by electrodepositing semiconductors in the nanometer-sized pores of anodic alumina films [55]. Current–voltage characteristics of the QDs showed Coulomb blockade at room temperature, while the wires also showed a Coulomb staircase when exposed to infrared radiation. These results indicate that the electrochemical self-assembly method have potential uses in room-temperature single electronics. The electrochemical self-assembly consists of three steps: (1) Electropolishing the aluminum foil in a suitable electrolyte to clean and prepare the surface; (2) anodizing the electropolished foil in either sulfuric or oxalic acid with a dc current to form a porous alumina film on the surface; (3) electrodepositing the material of interest within the pores. 

Stefanita et al. discovered that quasi-periodic arrays of QDs and nanowires of a wide variety of materials could be self-assembled using simple beaker electrochemistry in 2004 [56]. Furthermore, they described a room-temperature infrared photodetector that may have application in biologically inspired computing architectures. In 2006, Wang et al. reported the electric-field-modulated infrared absorption of CdS QDs by electrochemical self-assembly at room temperature [57]. The electrochemical self-assembly was realized by the electrodeposition of a semiconductor into a gap in an anodic aluminum oxide film. An electric field modulates this absorption by altering the overlap between the wave functions of electronic states in the QDs and the trap states in the surrounding alumina, thereby affecting the matrix element for radiative transitions, similar to the quantum confined Stark or Franz–Keldysh effect. The absorption is associated with the photo-assisted real space transfer of electrons from the CdS QDs to the surrounding trap sites in the alumina. 

Recently, Luo et al. again achieved the self-assembly of CdS QDs by electrochemically generated metal-ion crosslinkers in 2021 [58,59]. The self-assembly process used in situ generated metal ion crosslinkers (Ni^2+^, Co^2+^, Ag^+^, or Zn^2+^) within a colloidal solution of metal chalcogenide nanoparticles capped with ligands featuring pendant carboxylate groups. First, metal ion-mediated electrogelation–gelation of CdS QDs was initiated by applying a potential of 0.5 V vs. Ag/AgCl/sat. KCl at a Ni wire electrode immersed in the CdS QD solution. Then, a thin layer of the CdS QD gel was synthesized on the Ni anode surface after 15 min. Finally, the CdS QD gel growth continued for 1 h. Moreover, this method is amenable to the use of other metals with low oxidation potentials such as Co, Ag, and Zn, as shown in Figure 9b–e. The different QDs gel sizes were caused by the different electro-dissolution rates of these electrodes. In November of the same year, they further revealed competition between the two electrogelation pathways under the condition of high oxidation potentials and non-noble metal electrodes (including Ni, Co, Zn, and Ag). Relative to conventional chemical approaches, electrochemical self-assembly methods on QDs enable the use of two additional levers for tuning the assembly process: electrode material and potential. Therefore, they studied the self-assembly of QDs at different potentials in Figure 9f,g. The results show that QD gel formation was observed for the Co electrode at both 1 V and 2 V. After increasing the voltage, the metal ion/QD ratios also increased.

Researchers have shown that the QDs exhibited many excellent properties by the electrochemical self-assembly method. In 2014, Favaro et al. reported a bottom–up self-assembly strategy by the electrochemical method that exploits the self-assembly of the graphene oxide QDs produced during electrochemical erosion [60]. Moreover, the luminescent nanospheres were prepared by the self-assembly of the graphene oxide QDs around the frozen water nuclei. Compared to the use of conventional micrometric graphene sheets, the advantages of this electrochemical self-assembly approach reside not only in the possibility to downscale the control over the spatial organization, but also in the introduction of the intrinsic luminescent properties of the QDs in the final material, as shown in Figure 10a,b. 

In 2020, Luo et al. first proposed the electrochemical gel method, which uses electrochemical methods to self-assemble colloidal QDs into three-dimensional mesoporous structure aerosols [61]. This approach produces macroscale 3-D connected pore-matter nanoarchitectures that remain quantum confined and in which each QD is accessible to the ambient. For the case of CdS QDs, this is achieved by the electrochemical removal of surface-bound thiolate ligand “protecting groups” as dithiolates and the solvation of Cd ions, followed by oxidation of the exposed “core” chalcogenides to form interparticle dichalcogenide bonds. The redox-active nature of interparticle dichalcogenide bonds enables electrochemical disassembly of the gel network by reducing the dichalcogenide bonds to chalcogenides at negative potentials. To initiate this oxidative electrogelation, the electrode potential should be more positive than the oxidation potentials of thiolate ligands and the core chalcogenide. In Figure 10c, the self-assembled CdS QD gel sensor showed fast response–recovery dynamics at room temperature. In addition, the CdS gel sensor had the best combination of low limit of detection (11 ppb) and rapid recovery time (28 s) at room temperature, as shown in Figure 10d. More importantly, the various metal chalcogenide QDs (CdS, ZnS, and CdSe) can be assembled into macroscale 3-D connected pore-matter nanoarchitectures by this universality of strategy. 

Based on the above results, it can be seen that the electrochemical self-assembly methods have many advantages. First, the electrochemical self-assembly method is very simple, and only needs a beaker, electrode, power, and solution to be carried out smoothly. Second, compared to other self-assembly methods, the periodicity, size control, cost, and achievable nanostructure density of QDs can be more easily adjusted by electrochemical self-assembly. Third, because of the redox-active nature of the bonding between QD building blocks, the electrochemical self-assembly process is reversible. However, the types of QDs suitable for electrochemical self-assembly methods are limited. At present, electrochemical methods are mainly aimed at the self-assembly of chalcogenide QDs (CdS, ZnS, and HgS). Therefore, the application scope of the electrochemical self-assembly method remains to be broadened.

### 2.5. Chemical Vapor Deposition Method (CVD)

CVD is also an important method for the growth of nano-materials. Due to its low cost, easy operation, high yield, and large-scale production, it has attracted extensive attention from researchers. CVD is a technology in which materials produce physical and chemical phenomena under high temperature conditions, and the gaseous precursor reactants are dissociated in the high temperature deposition area, and then transported to the deposition location by carrier gas, and the solid film is generated by the chemical reaction between atoms and molecules.

The growth process of CVD mainly follows two growth mechanisms: one is the S–K growth mechanism of MBE above-mentioned, and the other is the gas–liquid–solid growth mechanism. The S–K growth mechanism has been described in detail above, and S–K growth occurs almost exclusively during lattice mismatched heteroepitaxy. The gas–liquid–solid growth mechanism is mainly introduced here. The growth mechanism is that metal or semiconductor liquid droplets condensing from the vapor phase catalyze whisker or fiber growth under high-temperature chemical vapor deposition conditions. The nanostructures synthesized by CVD are dependent on the synthesis time, and different nanostructures can be obtained at different times (Figure 11a) [62]. Meanwhile, taking ZnO as an example, we plotted the synthesis process of QD thin films by chemical vapor deposition (Figure 11b).

In 2002, Bell et al. found that the VLS method has obvious advantages for the growth of nitride QDs, because Ga is a liquid at 30 °C, it can be deposited into droplets, and its size can be fine-tuned by simple annealing at relatively low temperatures [63]. The growth of self-assembled gallium nitride QDs by the gas–liquid–solid mechanism was achieved for the first time. Furthermore, the perfectly coherent nanometer-sized GaN QDs with the wurtzite structure were demonstrated by high resolution XTEM images.

In the same year, Kim et al. proposed the use of a different oxygen source, nitrogen dioxide (NO_2_), to obtain self-organized ZnO QDs with smaller sizes and higher densities capable of resulting in apparent quantum size effects grown on SiO_2_/Si substrates by MOCVD [64]. The self-organized ZnO QDs in their work were grown on thermally formed SiO_2_ layers with a thickness of 25 nm on Si substrates by MOCVD, where diethylzinc as a zinc source and reactive NO_2_ gas as an oxygen source were used. The AFM images show that the size and density of ZnO QDs grown by self-assembly can be controlled by the growth conditions. At the same time, the structure and optical properties of QDs can be improved by post-growth annealing.

In 2005, Tan et al. prepared zinc oxide QDs on a silicon substrate by metal–organic CVD [65]. A large number of precursors were introduced in the growth process, and the zinc oxide QDs were embedded into the zinc oxide thin film through the strong reaction of the precursors. The photoluminescence of the thin film was detected at 80 K, indicating that a relatively uniform thin film was prepared by CVD. 

In 2013, Fan et al. demonstrated the fast CVD synthesis (in seconds) of graphene QDs (CGQDs) by carefully engineering the CVD parameters [66]. They explored the specific effects of the growth parameters including the temperature, flow rates of the carbon source and hydrogen, and the morphology of copper substrates on the size of CGQD during the preparation of graphene by chemical vapor deposition. They also found that CGQDs could easily be formed on copper substrates if the growth parameters are modified in a manner contrary to the strategies taken in a typical CVD process for the large domain growth of graphene. The CGQDs thus produced have a large area and controllable size, are highly dispersed, and can be transferred to arbitrary substrates while maintaining their pristine configuration. The Raman spectra of CGQDs are shown in Figure 11c. Figure 11d,e shows the CGQDs imaged with high-resolution scanning electron microscopy (SEM) observation (10 k magnification) and transmission electron microscopy (TEM). As shown in Figure 11e, the CGQDs they produced had excellent crystallinity.

In 2016, Yan et al. synthesized fluorescent carbon quantum dots (C-CQDs) with a graphite structure by CVD for the first time [67]. They found that the synthesized C-CQDs exhibited an excellent crystalline graphitic nature. Polymer solar cells (PSCs) were fabricated using C-CQDs as electron transport layer (ETL). Figure 12a,b shows the structure of the PSCs and the energy level diagram of various function layers. Figure 12c,d depicts the J–V curves and EQE spectra of P3HT:PC_61_BM devices. C-CQDs had a better interfacial connection with the polymer, allowing the device to have better electron injection and reduced interface resistance. The device based on C-CQDs showed better device performance than that based on lithium fluoride. More importantly, the thermal stability of the device based on C-CQDs was improved.

In 2018, Bi et al. studied this in depth. InN QDs were self-assembled by the MOCVD method and fabricated into nanowires [68]. InN QDs were formed on an InN wetting layer in S–K mode. By controlling the adatom migration, the InN QDs can be controlled to nucleate either only at the edges between the plane side facets or also on the side facets. Then, by controlling the length of adatom migration, the nucleation growth of InN QDs at the edge of the substrate can be controlled.

The film obtained CVD had the advantages of compact structure, uniform thickness, and good adhesion to the matrix. CVD has the advantages of simple equipment, simple operation, low cost, and high yield. During the experiment, the shape and quantity of the synthetic materials can be controlled by changing the temperature, pressure, carrier gas flow rate, and reaction time. In addition, the selection of the precursor, the distance between reaction sources, the treatment method and type of substrate, heating temperature, and air pressure are the main factors affecting the growth of CVD. However, at high temperature, the material vapor is not controllable, the component is single, the growth process cannot be monitored and there are other difficulties such as it cannot grow multi-component material heterojunction. At the same time, during the growth process of QD film, the temperature is much lower than that of the normal synthesis of QDs, so the material cannot be completely vaporized, resulting in the poor quality of the prepared QD film, which leads to great challenges in the growth direction of the sample with the traditional CVD method.

### 2.6. Ligand Exchange Method

QDs are typically prepared using solution-phase methods because these enable exquisite control over the size, shape, and composition, in large part aided by the presence of ligands that passivate the surface of the particles as they form from monomer precursors. However, the common long-chain organic moieties exploited for surface ligation necessarily restrict the interparticle interactions as well as communication with the ambient. The self-assembly of QDs by ligand exchange can effectively change the length of the chains. At the same time, the self-assembly process gives QDs a higher order. Therefore, the performance of QDs can be optimized. The self-assembly of QDs have been realized by the exchange of the native ligands with short-chain organics (ethanedithiol, aniline) or inorganic ions (halides, chalcogenidometallates) either before or after film deposition (typically by dip or spincoating), but the relatively high electronic resistivity remains a limiting factor in device performance. Based on the relative hydrophobicity of the surface ligands, the QDs were also seen to facilitate or inhibit assembly in organic solvents, which ultimately dictated the solubility of the hybrid monomer unit. Increasing the size of the QDs led to large hybrid sheets with regions of highly ordered square-packed QDs. A second, smaller QDs species can also be integrated to create binary hybrid lattices. 

Yang et al. synthesized the CdTe QDs with highly stable and tunable photoluminescence by self-assembly in 2015 [69]. After ligand exchange, CdTe QDs were assembled into a chain by controlling the hydrolysis and condensation reaction of 3-mercaptopropyl-trimethoxysilane, as shown in Figure 13b,c. The chain was then coated with a SiO_2_ shell from tetraethylorthosilicate. Moreover, the CdSe/ZnS QDs in these SiO_2_ spheres exhibited higher stability and retained their initial PL properties after self-assembly. In addition, the ligand exchange self-assembly method can also increase the absorption efficiency of QDs, together with their electronic and excitonic coupling, to enhance the charge carrier mobility.

In 2018, Chen et al. reported a series of research on the self-assembly of mercury chalcogenide colloidal QDs by ligand exchange [70]. First, the self-assembly of HgTe QDs was researched. In Figure 13d, a TEM micrograph showed that HgTe QDs were self-assembled into superlattices upon slow evaporation from tetrachloroethylene. The small size and shape variation of these particles were further demonstrated by their propensity to crystallize into well-ordered superlattices. Ultimately, HgTe QDs self-organize into face-centered cubic arrays with ordered domains of several square microns. In the same year, they successfully synthesized HgTe colloidal QDs with high monodispersity and further regulated the doping state of HgTe QDs by chemical and electrochemical methods. In addition, direct spectral evidence for the fine structure of the conduction band in electron-doped HgTe QDs was found.

In 2020, researchers demonstrated a general and scalable approach to increase both light absorption and excitonic coupling of QDs by self-assembly [71]. QDs are self-assembled into colloidal superlattices using an emulsion template by ligand exchange. These colloidal superlattices exhibit extended resonant optical behavior, resulting in an enhancement in absorption efficiency in the visible range of more than two orders of magnitude with respect to the case of dispersed QDs. Zheng et al. proposed a self-assembly passivation strategy of QDs that was introduced to improve QD coupling in 2021 [72]. The self-assembly method utilizes ligand exchange using ammonium iodide, which could efficiently replace the OA ligands with I^−^ and the QDs inks can be highly stabilized due to a good charge balance on the QD surface. Then, the PbI_2_ passivation layer was assembled on the QD surface forming QD-PbI_2_, as shown in Figure 13e. The self-assembled PbI_2_ layer not only efficiently passivated the QD surface, but also worked as a “charge bridge” between adjacent QDs to improve the charge transport. The solar cells based on self-assembly QDs gave an improved power conversion efficiency of 12.3%. This work provides a simple and facile way to improve the electronic coupling of QDs.

In 2022, Coropceanu et al. reversibly self-assembled colloidal nanocrystals of gold, platinum, nickel, lead sulfide, and lead selenide with conductive inorganic ligands into superlattices, exhibiting optical and electronic properties consistent with strong electronic coupling between the constituent nanocrystals [73]. The results overcome the limitation of insulating organic surface ligands on the collective electronic states in ordered nanocrystal assemblies. In the same year, researchers developed a one-step ligand-exchange method to produce QD-DNA conjugates from commonly available hydrophobic QDs, as shown in Figure 13f,g. The results demonstrate that these bioconjugates have sufficient colloidal stability for DNA-directed self-assembly.


Figure 13(**a**) Schematic illustration of the process of the self-assembled QDs by the ligand exchange method [74]. Copyright 2017, *Journal Of The American Chemical Society*. (**b**,**c**) TEM images of CdTe QD assembly in low magnification and high magnification [69]. Copyright 2015, *Materials Research Bulletin*. (**d**) TEM micrograph shows the HgTe QDs self-assembled by the ligand exchange method. The inset shows a monolayer of the same particles drop-cast from a dilute solution [70]. Copyright 2018, *ACS Nano*. (**e**) HRTEM image of the CQD-PbI_2_. The PbI_2_ passivation layer was assembled between the CQDs [71]. Copyright 2021, *Advanced Materials Interfaces*. (**f**) Assembly results for 5 nm PbS QDs flocculated with K_3_AsS_4_ followed by the fast addition of ACN. (**g**) Assembly results for 6 nm PbSe QDs stabilized by K_4_Sn_2_S_6_ ligands in N-methylformamide [73]. Copyright 2022, *Science*.
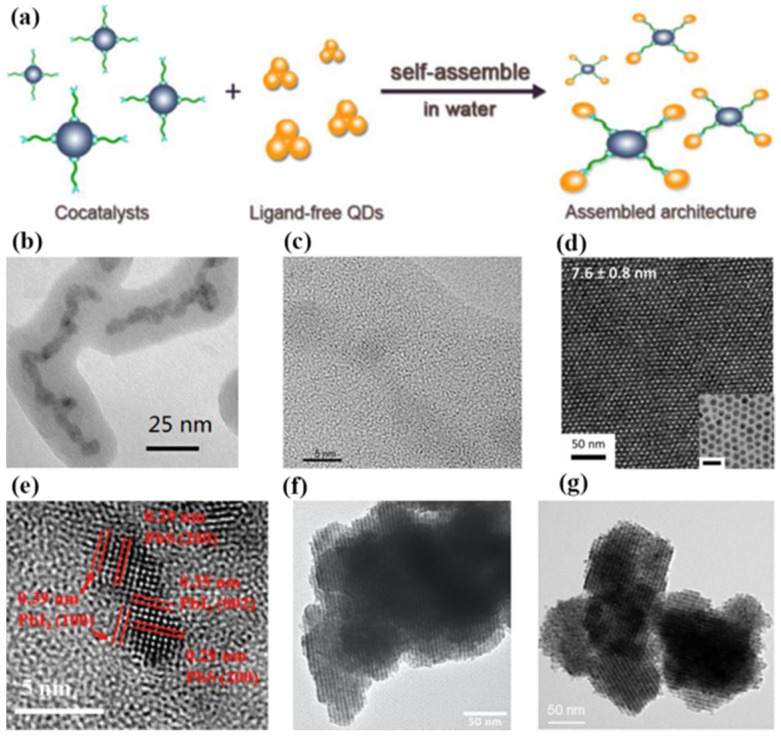



The controllable and effective interaction of colloidal QDs with cocatalysts is hindered by their ultra-small size. In order to break through this bottleneck, Li et al. initiated a ligand exchange self-assembly method to fabricate assembled frameworks of QDs and Pt nanoparticles for facile and ultra-fast interparticle electron transfer in 2017 [74]. In this contribution, the surface ligands, polyacrylate anions, were used to strongly bind light absorbers (QDs) and cocatalysts (Pt-nanoparticles) together to form the advanced architecture. The rate and efficiency of interfacial electron transfer were significantly enhanced after self-assembly. As shown in Figure 14a, the assembled CdSe/CdS QD/cocatalyst solution produced 4183 ± 67 μmol of molecular H_2_ in 8 h, which was significantly higher than the QDs without self-assembly. Moreover, the assembled CdSe/CdS QD/cocatalyst solution exhibited a highly internal quantum yield of 65% and a total turnover number of >1.64 ×10^7^ per Pt nanoparticle. These results demonstrate that the ligand exchange self-assembly method is a promising way to improve the sluggish kinetics of the interparticle electron transfer process.

In 2019, Chen et al. effectively improved the carrier mobility via self-assembled HgTe colloidal QDs [75]. By adjusting the concentration of mercuric chloride in the process of ligand exchange, the doping state of HgTe QDs was effectively regulated, and the precise regulation of HgTe QDs from p-type doping to n-type doping was realized. After mixed ligand exchange and polar phase transfer, infrared photoconductor performances based on colloidal HgTe QDs showed an order of magnitude enhancement. For example, the responsivity and detectivity of the detector increased by 100 and 10 times, respectively.

Later, they also achieved the self-assembly of thin-film HgSe colloidal QDs by ligand exchange in polar inks, thereby significantly improving the electron and hole mobility of HgSe QDs [76]. Studies have shown that the mobility of HgSe QDs with a diameter of 7.5 nm is as high as about 1 cm^2^/Vs, as shown in Figure 14b. In the same year, they designed a HgTe QDs film with higher mobility based on self-assembly characteristics [77]. Through discrete QD state transmission, the QD film showed a high charge mobility (8 cm^2^/Vs), as shown in Figure 14c. On this basis, they effectively eliminated the surface state and achieved air-stable n and p doping regulation by the hybrid surface passivation process. This study solved important problems about the delocalization and hopping mechanism of transmission in QD solids and provides the possibility for improving QD technology.

Samaneh et al. used organic–inorganic perovskites with mixed halides for the solid-state exchange of oleic acid ligands on PbS QDs in 2021 [78]. A compact arrangement of the nanocrystals was procured in a thin film (~100 nm) via self-assembling. The oleic acid was replaced with FA by immersion (2 min) of the films in the solution. Transmission electron microscopy showed that nano-scale cracks, short-range ordering, and fusion of the nanocrystals occurred during the exchange with FAPbI_3_. Gradual substitution of I^-^ with Br^-^ ions improved the homogeneity of the film with fewer defects. Furthermore, uniform cubic assembly of the nanocrystals without defects and high coverage density were obtained with FAPbBr_3_. The results indicate that the self-assembled PbS QDs exhibited better performance than the randomly distributed PbS QDs by ligand exchange methods. Figure 14d,e shows the self-assembled PbS QDs with low threshold voltage, uniformly distributed SET voltages, fast response time, and high resistance [79]. These excellent performances were attributed to the ordered arrangement of the PbS QDs in the self-assembled process. Moreover, biosynaptic functions and plasticity were implemented successfully in the memristor device with the self-assembled PbS QDs.

## 3. Conclusions

In this review, we summarized the existing methods of the self-assembly of QDs. We mainly introduced the development of the molecular beam epitaxy method, gravity sedimentation method, colloidal solution method, electrochemical method, chemical vapor deposition method, ligand exchange method, and some novel self-assembly methods of QDs. The self-assembly process and applicable QD system of each method were described in detail. Moreover, we summarized the performance optimization of QDs by various self-assembly methods. The self-assembly process can endow QDs with an adjustable size. At the same time, the stability and mobility of self-assembled QDs are effectively improved. In addition, the self-assembly method can suppress the interference of heat flow and non-radiative transitions on QDs. Finally, we propose the problems and possible solutions of these self-assembly methods on QDs.

## 4. Challenges and Prospect

The self-assembly method has the advantages of simplicity and low cost. At the present stage, while the self-assembly methods show great prospects for optimizing the performance of QDs, they still face major challenges not only in material optimization, but also in device modification. For example, the application scope of the self-assembly method is limited. Self-assembly can only be carried out under certain conditions. For example, the types of QDs suitable for the electrochemical self-assembly method are limited. Currently, the electrochemical methods are mainly aimed at the self-assembly of chalcogenide QDs. Furthermore, the long-range order of QD solids is difficult to control. Defects are easily found in self-assembled QD solids. Combining multiple self-assembly methods to achieve a versatile self-assembly method on QDs might prove to be a good solution.

## Figures and Tables

**Figure 1 materials-16-01317-f001:**
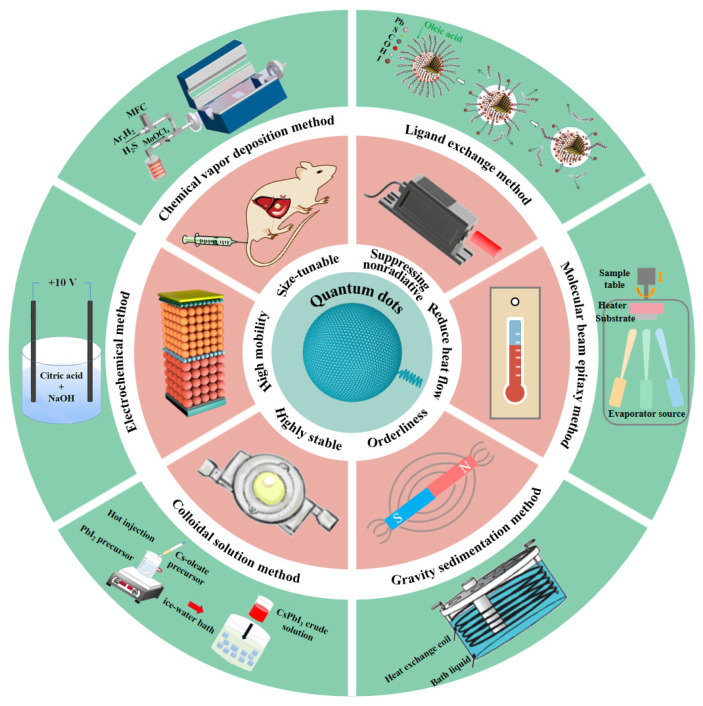
The performance optimization and methods of self-assembled QDs.

**Figure 2 materials-16-01317-f002:**
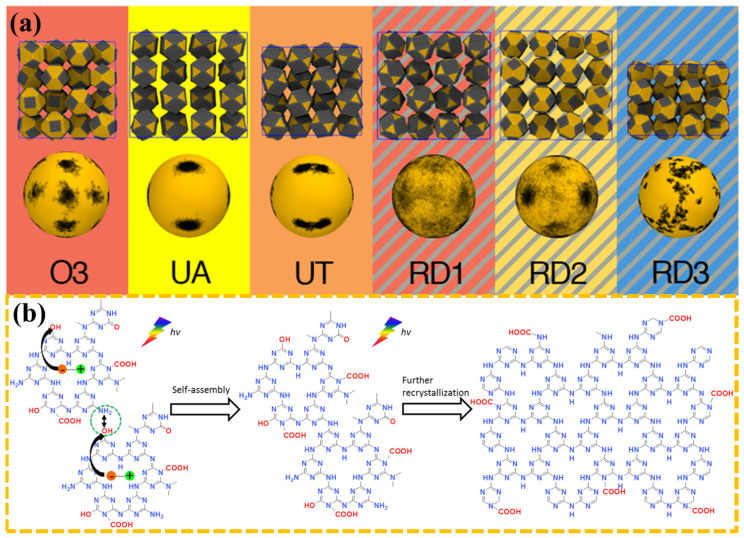
(**a**) Images of various QDs after molecular dynamics simulation [22]. Copyright 2019, *J Am Chem Soc*. (**b**) A schematic diagram of light-induced self-assembly of silicon nitride [23]. Copyright 2021, *Angew Chem Int Ed Engl*.

**Figure 4 materials-16-01317-f004:**
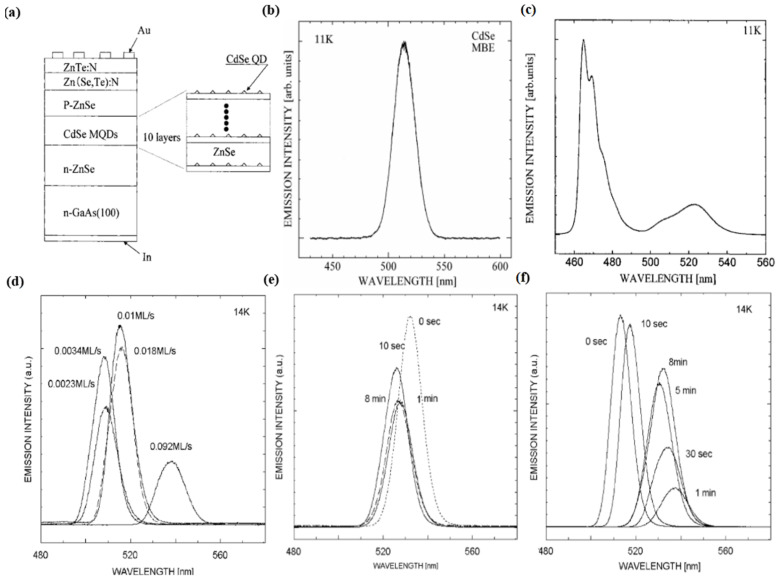
(**a**) Structure of the ZnSe p-n diode with CdSe multi QDs layers [35]. Copyright 2002, *Physica status solidi*. (**b**) PL spectrum of CdSe QDs fabricated by MBE [36]. Copyright 2000, *Japanese Journal of Applied Physics*. (**c**) Structure of ZnSe p-n diode with CdSe multi QDs layers [35]. Copyright 2002, *Physica status solidi*. (**d**) Dependence of the photoluminescence spectra on the rate of the CdSe source supply. (**e**,**f**) Dependence of the photoluminescence spectra on the growth interruption time between the growth of the CdSe QDs layer and the ZnSe intermediate layer at a source supply rate of (**e**) 0.01 and (**f**) 0.1 ML/s [37]. Copyright 2007, *Journal of Crystal Growth*.

**Figure 5 materials-16-01317-f005:**
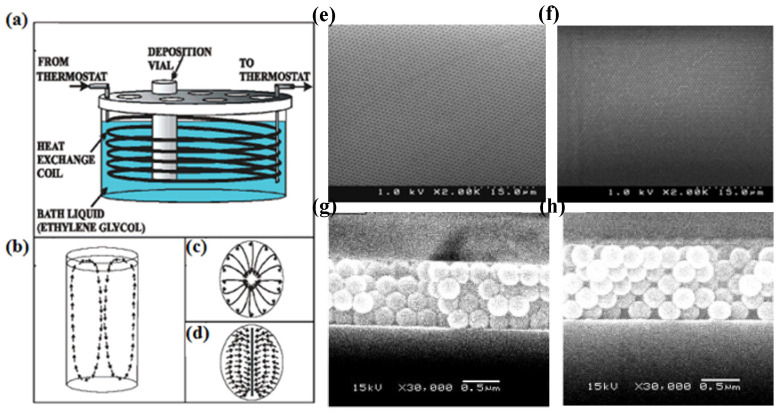
(**a**) Diagram of the isothermal bath device for the deposition of colloidal crystal film. (**b**–**d**) Representation of a convection pattern in a vial immersed in a bathtub. (**b**) Side view, (**c**) top view, and (**d**) top view when the slide is inserted into the center of the vial. (**e**,**f**) SEM image of a top view of the silica gel film, which shows that the prepared film achieved a high degree of sequencing. (**e**) 480 nm, (**f**) 850 nm [43]. Copyright 2003, *Journal of the American Chemical Society*. (**g**,**h**) Cross-sectional SEM images of polystyrene. (**g**) −10°, (**h**) 0° [44]. Copyright 2003, *Chemistry of Materials*.

**Figure 6 materials-16-01317-f006:**
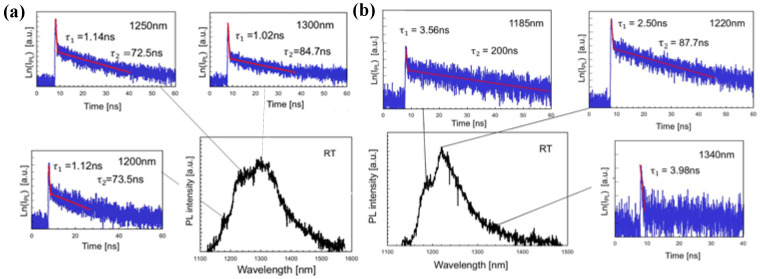
(**a**) μ-PL spectra and decay curves of the QD films prepared for 20 min on a flat substrate. The vertical axes of the decay curves are the natural logarithm of PL intensity. (**b**) μ-PL spectra and decay curves of the QDs films prepared for 7 d on a template. The vertical axes of the decay curves are the natural logarithm of PL intensity [47]. Copyright 2020, *Japanese Journal of Applied Physics*.

**Figure 7 materials-16-01317-f007:**
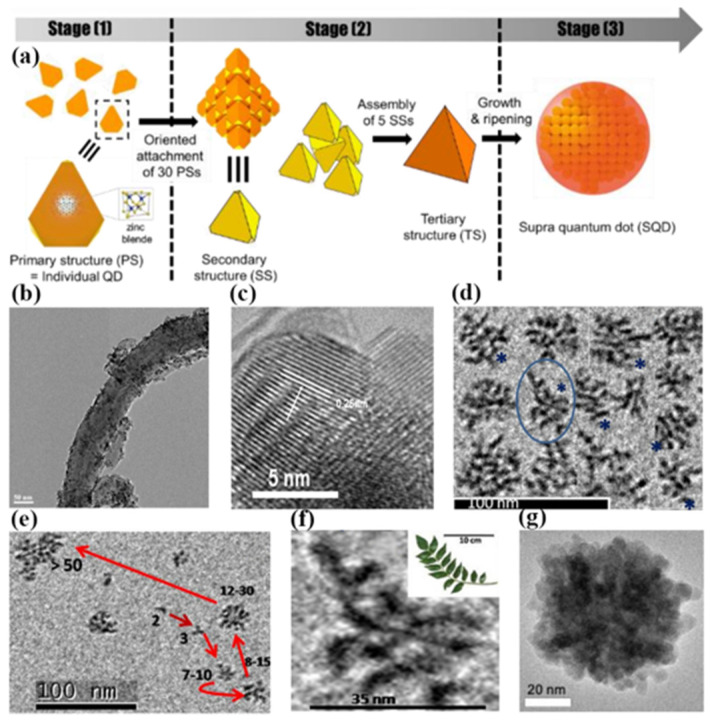
(**a**) Schematic illustration of different growth stages of SQDs with the assembly mechanisms. (**b**) TEM image of a single multi-walled carbon nanotubes coupling with ZnO QDs. (**c**) HRTEM image of ZnO QDs beaded on the sidewalls of multi-walled carbon nanotubes [48]. Copyright 2009, *Carbon*. (**d**) TEM image of assemblies of ZnO QDs into various dendritic nanostructures (* represents 1D nanostructures). (**e**) TEM image illustrating process of the evolution of large dendritic nanostructure from drop shape QDs (Arrows illustrate the process of evolution of large dendritic nanostructure from drop shape QDs.) and (**f**) enlarged image of a single ZnO dendron [49]. Copyright 2012, *Applied Surface Science.* (**g**) TEM image of CdSe SQDs [50]. Copyright 2016, *Chemistry of Materials*.

**Figure 8 materials-16-01317-f008:**
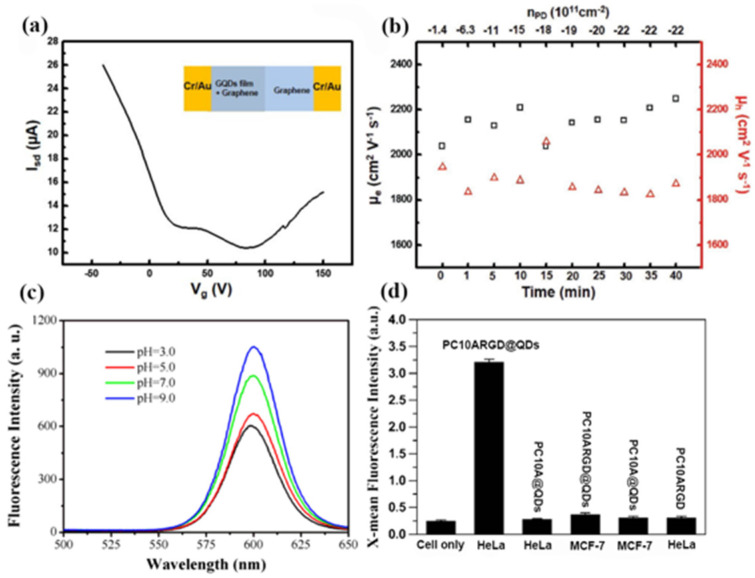
(**a**) Isd–Vg curve of graphene covering both of the partially deposited graphene QD film and SiO_2_ device. (**b**) FET electron mobilities and hole mobilities of graphene/graphene QD film with respect to the increase in photoinduced doping (The triangles in the figure represent the mobility of the electron and the squares in the figure represent the mobility of hole.) [51]. Copyright 2017, *Small*. (**c**) Fluorescence spectra of PC10ARGD@QDs hybrid nanogels at different pH from 3 to 9. (**d**) Flow cytometry of HeLa and MCF-7 cells incubated with PC10A@QDs, PC10ARGD@QDs hybrid nanogels, and free PC10ARGD polypeptide [52]. Copyright 2017, *Journal of Nanoparticle Research*.

**Figure 9 materials-16-01317-f009:**
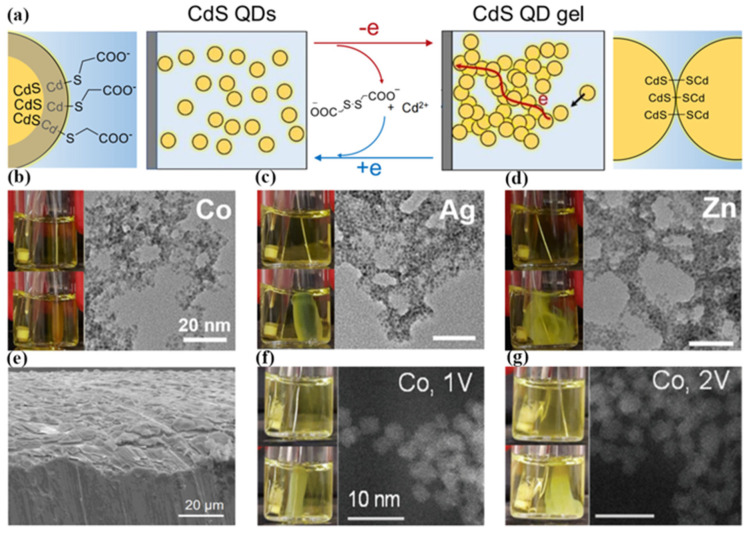
(**a**) Schematic diagram of QD self-assembly by the electrochemical method. (**b**–**d**) Metal ion-mediated electrogelation-gelation of CdS QDs using different metal anodes Co, Ag, and Zn, respectively. (**e**) SEM image of the CdS QDs gels grown on a Ni foil electrode surface [58]. Copyright 2021, *Chemistry of Materials.* (**f**,**g**) Photographs and STEM micrographs of the QDs gels produced using Co electrodes at 1 V and 2 V [59]. Copyright 2021, *Nanoscale*.

**Figure 10 materials-16-01317-f010:**
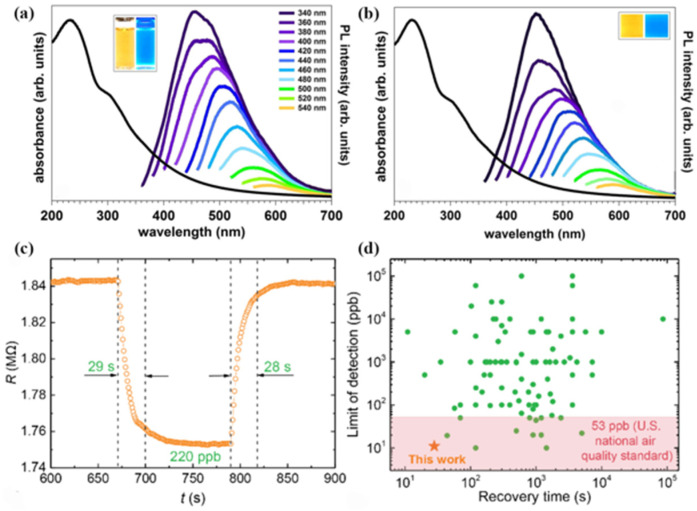
UV–Visible absorption spectra and photoluminescence measurements of (**a**) as-prepared graphene oxide QDs obtained from electrochemical etching in phosphate buffer solution and (**b**) graphene oxide QD-based nanospheres [60]. Copyright 2014, *Carbon.* (**c**) Response–recovery curve of the CdS gel sensor in the presence of 220 ppb of NO_2_. (**d**) Limit of detection and recovery time of CdS gel sensor [61]. Copyright 2020, *Journal of the American Chemical Society*.

**Figure 11 materials-16-01317-f011:**
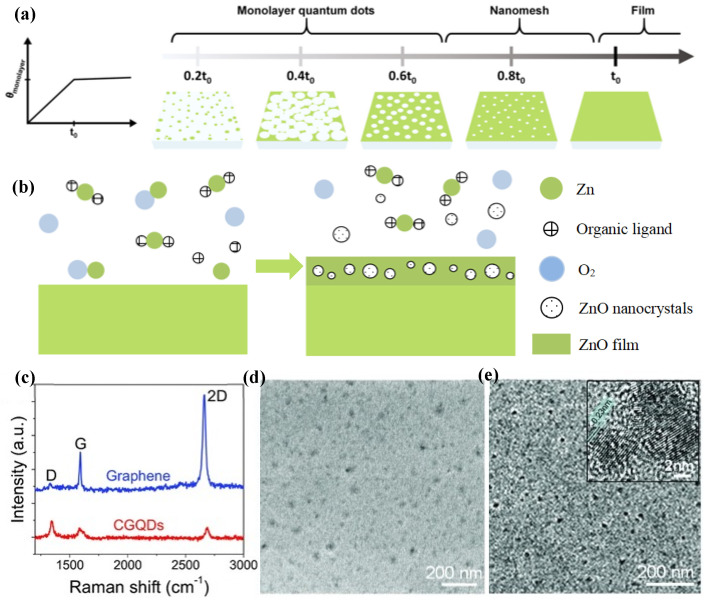
(**a**) Synthesis of the nanostructure of the nanomesh and QDs depending on the growth time [62]. Copyright 2022, *Bulletin of the Korean Chemical Society.* (**b**) Growth mechanism of the ZnO QD embedded film. Arriving of the precursors on the hot substrate. Pre-reaction of the precursors facilitates the formation of nanocrystals, and in turn incorporates into the ZnO film. (**c**–**e**) Structural characterizations of CGQDs. (**c**) Raman spectra. (**d**) SEM image. (**e**) TEM image. The inset shows the HRTEM image of two CGQDs [66]. Copyright 2013, *Particle & Particle Systems Characterization*.

**Figure 12 materials-16-01317-f012:**
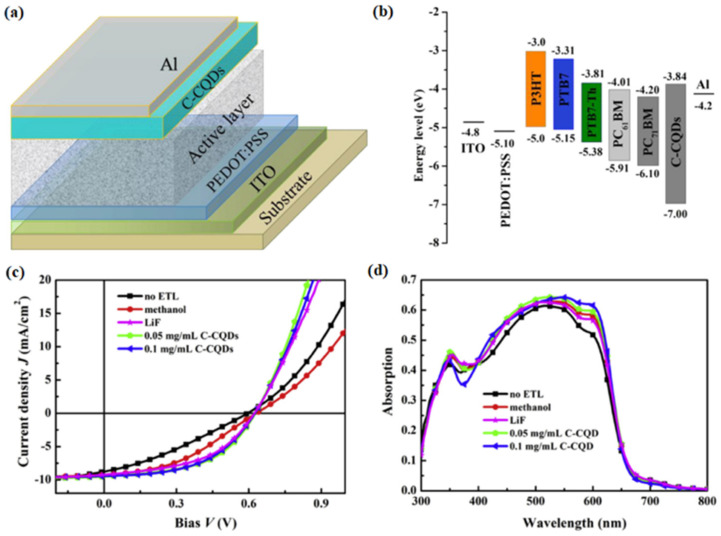
(**a**) The structure of the PSCs. (**b**) The energy level diagram of various function layers. (**c**) The J–V curves of P3HT:PC_61_BM devices. (**d**) The EQE spectra of P3HT:PC_61_BM devices [67]. Copyright 2016, *Carbon*.

**Figure 14 materials-16-01317-f014:**
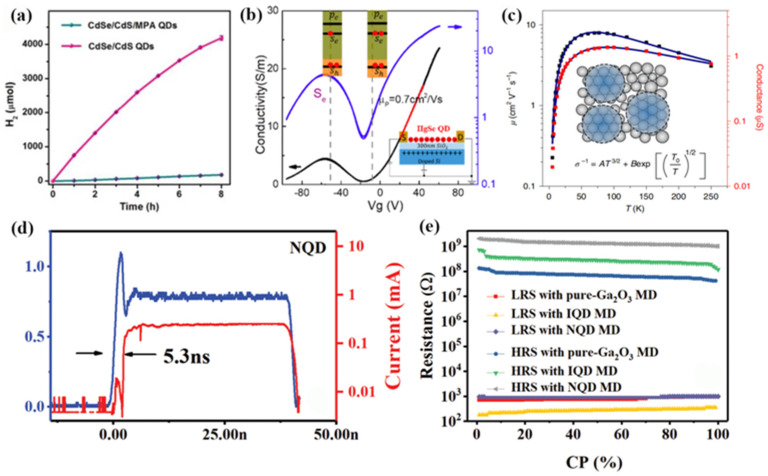
(**a**) Photocatalytic H_2_ evolution of CdSe/CdS QD/Pt nanoparticles with and without self-assembly [74]. Copyright 2017, *Journal Of The American Chemical Society*. (**b**) Field-effect transistor source-drain current of the HgSe hybrid QDs at 80 K [76]. Copyright 2020, *The Journal of Physical Chemistry Letters*. (**c**) Fitting images of temperature-dependent mobility and conductivity of HgTe QDs solids [77]. Copyright 2020, *Nat Mater*. (**d**) SET response time for PbS QDs memristor devices. (**e**) Statistic data of resistance distribution for sweeping over 60 I–V cycles [79]. Copyright 2019, *Adv Mater*.

**Table 1 materials-16-01317-t001:** Comparison of the Frank van der Merve, Volmer–Weber, and Stranski–Krastanow methods.

Mode	The Form of the Film	Applicable Condition (The Lattice Mismatch between Epitaxial Material and Substrate Material)	Applicable System
Frank van der Merve mode	Two-dimensional layered structure	Small	-
Volmer–Weber mode	Three-dimensional island structure	Large	InPSb/InP QD systems InN/GaN QD systems et al.
Stranski–Krastanow mode	First a two-dimensional layered structure, then a three-dimensional island structure	Between 5% and 10%	InGaAs/GaAs(001) QD systems et al.

## Data Availability

Not applicable.

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
