# Peer review of "Development of Self-Assembly Methods on Quantum Dots"

_materials, 2023, doi:10.3390/ma16031317_

Round 1

Reviewer 1 Report

Please find the reviewer comments attached herewith.

Reviewer 2 Report

Review Report

Manuscript title: Development of Self-assembly Methods on Quantum Dots

Manuscript ID: materials-2186827

Report: The review work is very interesting. This review describes about the development of Self-assembly methods on Quantum Dots. It is well organized. This would be published after the revision of the following issues.

1.       The introduction is described in very limit including only 6 references. It should go through details description with minimum 20-25 references for better improvement of the manuscript. Include the most recent refences.

2.        In section 2.1, make one table and compare the 3 different modes of the assembly methods.

3.       In Fig. 5, there is a very good comparison. It would be more better if you can arrange in the same micrometer of the SEM images.

4.       In Fig. 7 & 9, there are only HRTEM images of ZnO QDs & CdS, need to include also SEM images.

5.       Need to check the grammatical corrections of English.

Round 2

Reviewer 1 Report

Authors have satisfactorily addressed the reviewer comments.

Reviewer 2 Report

The authors tried to revise the manuscript with their best efforts and it has been improved. I recommend accepting it in its current form.